# The Longitudinal Relationship between Internet Addiction and Depressive Symptoms in Adolescents: A Random-Intercept Cross-Lagged Panel Model

**DOI:** 10.3390/ijerph182412869

**Published:** 2021-12-07

**Authors:** Xiaoyan Yi, Guangming Li

**Affiliations:** 1School of Psychology, South China Normal University, Guangzhou 510631, China; yxy2020@m.scnu.edu.cn; 2Key Laboratory of Brain, Cognition and Education Sciences, Ministry of Education, South China Normal University, Guangzhou 510631, China; 3Center for Studies of Psychological Application, School of Psychology, South China Normal University, Guangzhou 510631, China; 4Guangdong Key Laboratory of Mental Health and Cognitive Science, South China Normal University, Guangzhou 510631, China

**Keywords:** internet addiction, depressive symptoms, middle school students, RI-CLPM

## Abstract

Internet addiction and depressive symptoms are extremely common problems among teenagers, and the coping strategy has been proved to be closely related to internet addiction and depressive symptoms. Based on three waves of data from a sample of Chinese middle-school students (*N* = 1545, *M_age_* = 14.88 years old, *SD* = 1.81; 55.00% females), this study examines the longitudinal relationship between internet addiction and depressive symptoms among adolescents ultilizing the random-intercept cross-lagged panel model. Results revealed a unidirectional predictive effect of depressive symptoms at T_2_ on internet addiction at T_3_, but not vice versa, the effect was more significant in the male group. Positive coping strategies had a significant negative predictive effect on the random intercept of internet addiction and depressive symptoms, while negative coping style had a significant positive predictive effect on the random intercept of internet addiction and depressive symptoms. Effective identification and intervention of depressive symptoms may be beneficial to the intervention and prevention for internet addiction, and we should pay attention to the cultivation of middle school students’ positive coping strategies.

## 1. Introduction

Internet addiction and depressive symptoms are extremely common problems among teenagers today. The number of internet users in China had reached 1.01 billion, and youth aged 10–19 accounted for 12.3% of total internet users by June 2021 [1]. While the internet brings many conveniences to people’s lives, it also has some negative effects. Among them, the problem of teenagers’ internet addiction has attracted increasing attention from all walks of life. internet addiction (IA), also known as problematic internet use (PIU), is defined as an impulse-control disorder that does not involve intoxicants [2]. In 2020, the number of juvenile internet users in China was 183 million, and 19.6% of them believed that they were psychologically dependent on the internet [3] (CNNIC, 2020). internet addiction can lead to obvious social, behavioral and health damage, such as loneliness and anxiety [4,5], which seriously affects the healthy growth of adolescents.

In addition to internet addiction, depression is also very common among adolescents. Related studies show that the incidence of depressive symptoms in middle school students in China is between 25.5–44% [6]. Depression can affect children and adolescents’ development, school performance, relationship with peers and family, and even lead to suicide [7].

Coping strategy, as a relatively stable cognitive and behavioral style, has been proved to be closely related to internet addiction and depressive symptoms [8,9]. Exploring how coping strategy can affect internet addiction and depressive symptoms in one model will be helpful to understanding the relationship between the three variables in more depth and is of great significance for identifying potential risk groups.

### 1.1. Relationship between Internet Addiction and Depressive Symptoms

There are three hypotheses explaining the relationship between IA and depressive symptoms.

Kraut et al. proposed the social displacement hypothesis, believing that using the internet adversely affects social involvement and psychological well-being [10]. Addiction to online social interaction reduces the time teenagers spend interacting with friends and family in real life, which may lead to maladjustment and the emergence of negative emotions such as depression. Bryant and Zillmann proposed the mood enhancement hypothesis, suggested that individuals use media based on their emotions, which means that they will selectively contact online content according to their emotions [11]. The internet provides adolescents with social support, a sense of accomplishment and controls, allowing them to escape from the real world where emotional difficulties occur into the virtual world. Adolescents with depressive symptoms were more prone to use the internet to ease their depression, and therefore they are more likely than their peers to be addictive to the internet [12]. Davis (2001) proposed a cognitive-behavioral model of pathological internet use (PIU), positing that PIU results from problematic cognitions coupled with behaviors that either intensify or maintain the maladaptive response. Psychopathology (e.g., depression) is a distal necessary cause of symptoms of PIU. Depressive individuals may hold more pessimistic perceptions of their social skills and prefer online communications, so they may spend an excessive amount of time on the internet to seek support and pleasure, which may increase their risk of becoming problematic internet users [13]. 

Many studies have found that relationships between internet addiction and depressive symptoms among adolescents. In cross-sectional studies, some suggest that internet addiction may be a risk factor for depression [10,14], while some other studies suggest that depression may be a risk factor for internet addiction [12,15]. Although there are several longitudinal research works on the longitudinal relationship between internet addiction and depressive symptoms, the results are inconsistent. Some studies believe that there is only a unidirectional prediction relationship between the two variables [16,17], and some that there is a bidirectional prediction relationship [18,19]. 

Most of longitudinal studies employ the traditional cross-lagged panel model (CLPM), a model that explores causality among variables by predicting the extent to which an individual will deviate from the group mean at a previous time point by another variable at the next time point. The biggest limitation of this model is that it mixes within-person effects and between-person effects, making the causal relationship between variables unclear. Within-person effects focus on how the changes in one variable are associated with the changes in another variable over time within an individual; between-person effects focus on how the changes in one variable are associated with the changes in another variable over time between different individuals. When CLPM is used to explore the longitudinal relationship between internet addiction and depressive symptoms, it is assumed that each individual has the same baseline level of internet addiction and depressive symptoms, meaning that all individuals’ scores fluctuate around the same baseline level (the group mean), and there is no stable individual differences. Obviously, this condition is hard to meet. Because of the baseline level of individual depressive symptoms, internet addiction is bound to show stable differences.

In order to capture the stability difference between individuals and separate the within-person effects and between-person effects, some scholars proposed the random intercept cross-lagged model (RI-CLPM) [20]. Therefore, the current study will also use this model to explore the longitudinal relationship between adolescent internet addiction and depressive symptoms.

### 1.2. Gender Differences in Internet Addiction 

A lot of cross-sectional studies found that there is a gender difference in adolescent internet addiction, and males are more prone to internet addiction than females [21,22]. Yu et al. conducted 3-year longitudinal research on Hong Kong adolescents, and found that male adolescents were 40% more likely to become addicted to the internet than female adolescents [23]. Many reports point to gender differences in motivation for using the internet. Men tended to experience more addictive behaviors when playing games related to power, control or explore sexual fantasies online, while women were more likely to communicate with close, anonymous friends online to share their feelings and emotions [2]. Men and women use the internet as a result of different patterns of behavior and motivation: compared with women, men are more easily depressed, avoid emotional problems, so more inclined to use internet addiction to avoid depression, the internet is more likely to be for fun, and tend to be with friends on the internet, so depression in men is predicted from subsequent internet addiction. However, women are more inclined to search for information on the internet and prefer to surf the internet alone, which makes women more prone to internet addiction due to overuse of the internet [19].

### 1.3. Gender Differences in Depressive Symptoms

There is a lot of evidence that gender differences in depression occur in early adolescence [24,25]. Before puberty, boys had higher levels of depression than girls, but after puberty, the results were reversed [26]. Women generally show increasingly pronounced symptoms of depression than men, and the gender difference becomes more pronounced during adolescence [27]. During adolescence, women face earlier biological changes and are more influenced by the quality and stress of the social environment, so they are more prone to developing depressive symptoms [28].

Above all, there are gender differences in adolescent internet addiction and depressive symptoms, but few researchers studied gender differences in longitudinal relationships between adolescent internet addiction and depressive symptoms by separating the between-person and within-person effect, which therefore still needs further research.

### 1.4. Coping Strategy and Internet Addiction, Depression 

Coping is the cognition and behavior that individuals use to evaluate or reduce stress caused by life events in order to relieve internal tension [29]. Coping style, also known as coping strategy or mechanism, refers to the cognitive and behavioral styles adopted by individuals in the face of setbacks and pressures. Coping strategies can be classified into positive coping strategies and negative coping strategies according to their impact on mental and physical health, the former is an positive, effective and adaptive coping strategies adopted by individuals; the latter is a negative, ineffective and maladaptive coping strategies adopted by individuals. 

Numerous studies have pointed out that coping strategy is closely related to internet addiction [30]. Whang found that compared with potential internet addicts and non-internet addicts, internet addicts reported more that they played online games because they wanted to escape from reality [31].

Individuals who tend to use a positive coping strategy tend to use information on the internet to solve problems in real life, then internet use is “problem solving” oriented, they are not prone to suffer from internet addiction. Al-gamal et al. found that individuals who chose a problem-solving coping strategy had a lower level of internet addiction [8].

Many researchers have found that an individual’s coping strategy can affect his or her mental health. In terms of coping with stressful situations, adolescents who adopt a negative coping strategy have more psychological problems than those who adopt a positive coping strategy [32,33]. While individuals use negative coping strategies their anxiety and depression levels also increase [9,34]; and the increase in depressive symptoms, anxiety and other mood problems is associated with a decrease in positive individual coping strategies [35].

In conclusion, coping strategies are closely related to internet addiction and depressive symptoms, but few studies have explored how coping strategies can affect internet addiction and depressive symptoms at between-person level, so it is necessary to investigate the influence of coping strategies on adolescent internet addiction and depressive symptoms at a between-person level so as to identify potential risk groups.

## 2. Methods

### 2.1. Participants and Procedure

Participants were teenagers (*N* = 1545, 55.00% females) aged between 12 and 18 (*M* = 14.88; *SD* = 1.81) from three middle schools in southern urban China. After excluding the subjects who did not take part in the measurement twice, 1459 valid subjects were retained. The reasons for the loss of subjects were as follows: (1) absence on the test day. (2) The questions were filled in wrong or missed. Table 1 presents the demographic information of the subjects.

We randomly selected three middle schools in urban areas in southern China, and then randomly chose 17 classes from these three middle schools. Data were collected using paper/pencil tests in classrooms every three months for a total study span of six months. Each participant had a unique ID, and the same ID was used at all three waves. No participants changed classes or schools during these six months. The measurement sites were provided by the Academic Affairs Offices of the three middle schools. Researchers were trained before they administered the survey. Both schools and parents agreed the assessment, and this research was approved by the university research ethics board (Institutional Review Board).

### 2.2. Measures

#### 2.2.1. Internet Addiction

The Chinese version of Young’s Internet Addiction Test (IAT) (Young, 1998) [2] was used to measure internet addiction. IAT has 20 items, all of them rated on five-point scales, ranging from 1 (rarely) to 5 (always) and summed, the higher the total score, the more serious level of internet addiction. The Cronbach’s alpha for those items were good (T_1*α*_ = 0.909; T_2*α*_ = 0.917; T_3*α*_ = 0.932).

#### 2.2.2. Depressive Symptoms

Depressive symptoms was assessed using Zung’s self-rating depression scale (SDS) [36]. SDS had 20 items (e.g., “I feel down-hearted and blue.”), all of them rated on five-point scales, ranging from 1 (a little of the time) to 5 (most of the time) and summed, the higher the total score, the more serious level of depressive symptoms. The Cronbach’s alpha for those items were good (T_1*α*_ = 0.795; T_2*α*_ = 0.827; T_3*α*_ = 0.850). 

#### 2.2.3. Coping Style

Coping styles was measured in the first wave using the Chinese version of the Trait Coping Style Scale (TCSQ). It has 20 items, 10 of them used to measure negative coping (NC) while the other 10 used to measure positive coping (PC); all of items are rated on a five-point scale, ranging from 1 (never) to 5 (always). The higher the score for each dimension, the more likely participants are to adopt this coping style. The Cronbach’s alpha for those items was good (T_1*α*_ = 0.857).

### 2.3. Statistical Analyses

Statistical analyses were conducted using SPSS26.0 (IBM Corporation, Armonk, NY, USA) and Mplus 7.4 (MPLUS 7.4 is a data analysis and statistical software developed by MUTHEN & MUTHEN located in Los Angeles, CA, USA). The evaluation of model fitting adopts the following indexes proposed by Kline: the chi-square statistic (*χ*^2^), the root-mean-square error of approximation (RMSEA; acceptable < 0.08, good < 0.05), the standardized root-mean-square residual (SRMR; acceptable < 0.08, good <0.05) and the comparative fit index (CFI; acceptable > 0.90, good > 0.95) [37].

Utilizing the RI-CLPM, we decomposed the within-person effects and between-person effects among associations between internet addiction (IA) and depressive symptoms. Next, we divided males and females into two groups and established the RI-CLPM, respectively, to examine gender differences in the relationship between internet addiction (IA) and depressive symptoms; we also included covariates (age and coping style) to investigate their effects on random intercepts of internet addiction and depressive symptoms.

## 3. Results

### 3.1. Descriptive Statistics, Correlations, and Prevalence of Internet Addiction (IA) and Depressive Symptoms

Descriptive statistics of all variables and prevalence of IA and depressive symptoms across waves are shown in Table 2. 

The correlations between IA and depressive symptoms are shown in Table 3. A significant positive correlation was found between all variables. 

### 3.2. The Results of the Random Intercept Cross-Lagged Model (RI-CLPM)

The RI-CLPM demonstrated a good fit to the data (*χ*^2^ = 145.275 *df* = 13, *p* < 0.001; CFI = 0.961; TLI = 0.901; RMSEA = 0.084, 90% CI [0.072, 0.096]; SRMR = 0.063).

Figure 1 shows the results of the RI-CLPM among total sample. 

Focusing on the cross-paths between two key variables, the coefficients for depressive symptoms at T_2_(WD2) to IA at T_3_(WN3) were significant (*β* = 0.162, *p* = 0.003), indicating that depressive symptoms at T_2_(WD2) predicted the increases in IA at T_3_(WN3). The coefficients for depressive symptoms at T_1_(WD1) to IA at T_2_(WN2) were non-significant (*β* = −0.032, *p* = 0.470). Both the coefficients for IA at T_1_(WN1) to depressive symptoms at T_2_(WD2) (*β* = 0.067, *p* = 0.146) and the coefficients for IA at T_2_(WN2) to depressive symptoms (WD3) at T_3_ (*β* = 0.029, *p* = 0.448) were non-significant. 

### 3.3. Gender Differences in the Relationship between Internet Addiction and Depressive Symptoms and the Effects of Covariates

The RI-CLPM for male group showed acceptable model fits, model fitting index are shown in Table 4, model diagram is shown in Figure 2.

Among the male group, the coefficients for depressive symptoms at T_2_(WD2) to IA at T_3_(WN3) were significant (*β* = 0.164, *p* = 0.018), indicating that depressive symptoms at T_2_(WD2) predicted the increases in IA at T_3_(WN3). Other cross-lagged coefficients were all non-significant.

Focusing on the effects of covariates (age and coping style) on random intercept of internet addiction and depressive symptoms, age did not significantly predict the random intercept of internet addiction, but significantly negatively predicted the random intercept of depressive symptoms (*β* = −0.156, *p* < 0.001), in other words, the older adolescents have lower baseline level of depression. Positive coping style significantly negatively predicted the random intercept of internet addiction and depression (*β* = −0.204, *p* < 0.001; *β* = −0.756, *p* < 0.001). This suggests that individuals who are more inclined to use positive coping strategy have lower baseline levels of internet addiction and depressive symptoms. Negative coping styles significantly positively predicted baseline levels of internet addiction and depressive symptoms (*β* = 0.510, *p* < 0.001; *β* = 0.497, *p* < 0.001). This suggests that individuals who are more inclined to use negative coping strategy tend to have higher baseline levels of internet addiction and depressive symptoms. 

The RI-CLPM for female group showed acceptable model fits, model fitting index are shown in Table 5, model diagram is shown in Figure 3.

As shown in Figure 3, among the female group, the cross-lagged coefficients from IA to depressive symptoms were not significant, while both the coefficients for depressive symptoms at T_1_(WD1) to IA at T_2_(WN2) (*β* = −0.115, *p* = 0.064) and the coefficients for depressive symptoms at T_2_(WD2) to IA at T_3_(WN3) (*β* = 0.136, *p* = 0.104) were marginally significant, which was inconsistent with the male group.

Focusing on the effects of covariates (age and coping style) on random intercept of internet addiction and depressive symptoms, age did not significantly predict the random intercept of internet addiction or depressive symptoms, and this also differs from the male group (in which age significantly predicts the random intercept of depression). Positive coping style significantly negatively predicted the random intercept of internet addiction and depressive symptoms (*β* = −0.188, *p* < 0.001; *β* = −0.573, *p* < 0.001), suggesting that individuals who are more inclined to use positive coping strategy have lower baseline levels of internet addiction and depressive symptoms. Negative coping styles significantly positively predicted baseline levels of internet addiction and depressive symptoms (*β* = 0.403, *p* < 0.001; *β* = 0.457, *p* < 0.001), it suggests that individuals who are more inclined to use negative coping strategy tend to have higher baseline levels of internet addiction and depressive symptoms. 

## 4. Discussion

The present research explored the longitudinal associations between internet addiction (IA) and depressive symptoms among adolescents. By utilizing the RI-CLPM, we disaggregated the within-person and between-person effects, and delineated the directionality of the association between two variables among total sample. Next, we established RI-CLPM in male and female group respectively to exam gender differences in the relationship between IA and depressive symptoms, and we also included covariates (age and coping strategies) to investigate their effects on random intercepts of internet addiction and depressive symptoms.

Our research found that among total group, depressive symptoms at T_2_(WD2) predicted the increases in IA at T_3_(WN3), other cross-lagged effects were not significant. It is obvious that our research supports the mood enhancement hypothesis, while two previous similar studies supported the social displacement hypothesis. Zhou et al. studied the longitudinal relationship between problematic internet use (PIU) and mental health among 1579 college students in China using the RI-CLPM [38]. They found that PIU predicted the increases in mental health issues over time instead of the reverse. Differences in findings between our study and Zhou’s study may be due to two reasons. 

First of all, college students have easier access to the internet and generally experience few parental restrictions [38], they also have more free time surfing the internet, so they are more easily addicted to the internet and thus this leads to mental health problems; while middle school students have a heavy academic burden, little free time and strict restrictions on internet use at schools. Therefore, it is unlikely that internet addiction will lead to mental health problems. On the contrary, they are more likely to resort to maladaptive strategies such as internet addiction to seek relief and comfort under a poor condition of mental health (e.g., depression).

Secondly, college students have higher population mobility, they tend to go to university across provinces or cities, away from the original local social network, in the face of new physical and interpersonal environment, and they are more prone to maladjustment. Whether the internet is used as a tool for keeping up with physically distant friends in their home town or as a tool for making new friends online, it is not conducive to their adaptation to the new real-world interpersonal environment, thus leading to loneliness and depression. Middle-school students usually study in the place of domiciliary registration, and rarely go to other provinces or cities. As a result, they are constantly in local social networks and often do not need the network to support strong ties. In addition, under the strict discipline of schools and home, they have few opportunities to make friends online, so they are less likely to break away from their existing strong relationships because they are too addicted to the internet.

Kojima et al. conducted RI-CLPM analysis to explore the relationship between PIU and depressive symptoms among junior high-school students in Japan [39]. They found that PIU predicted depressive symptoms. Differences in findings between our study and their study may be due to different sample characteristics (size, age, nationality and region) and measurements (source of information, depression scales and internet addiction scales). 

We also found gender differences in the relationship between internet addiction and depressive symptoms, the predictive effect of depression on internet addiction was more significant in the male group, so more importance should be attached to males. In addition, age in the male group significantly predicted the random intercept of depressive symptoms (i.e., the baseline level of the individual), while there was no such effect in the female group. Positive coping strategies had a significant negative predictive effect on the random intercept of internet addiction and depressive symptoms, while a negative coping style had a significant positive predictive effect on the random intercept of internet addiction and depressive symptoms, which was consistent in different gender groups. Therefore, we should pay attention to the cultivation of middle-school students’ positive coping strategies and reduce their tendency of using negative coping styles. 

According to the results of our research, adolescents who display a high level of depressive symptoms are in special need of attention, especially males. A screening program for depressive symptoms may provide the basis for effective intervention and prevention for internet addiction. At the same time, the cultivation of positive coping style will also help middle school students to relieve depression and reduce the baseline of internet addiction.

There are some limitations in the present research. Firstly, this study uses only self-reported questionnaires to collect data, although many previous studies on similar topics have done so, applying more diverse information collection methods will be better. Secondly, our participants are from only one city from southern China, this limited our findings’ generalizability. It is necessary to recruit participants from different areas and cultural backgrounds. Thirdly, although we found the predictive effect of depressive symptoms on internet addiction, we cannot over-interpret our findings, because the prediction of depressive symptoms to internet addiction just occurs in T_2_ to T_3_. This may be for the following reason. The prevalence rate of depressive symptoms at T_1_(T_1_ = 42.15%) is much higher than the other two time points (T_2_ = 31.53%, T_3_ = 37.01%), and this indicates that the number of people with depressive symptoms at this time point is far more than that of the other two time points. It is plausible that adolescents with mild depressive symptoms may have better emotional regulation ability and can relieve their depressive moods in other, healthier ways, and so they are less likely to become addicted to the internet, while adolescents with severe depressive symptoms cannot deal with their depressive moods well, so they would tend to resort to internet addiction more. The prevalence rate of depressive symptoms at T_2_ and T_3_ is similar, indicating that people with severe depressive symptoms stayed. Therefore, the depressive symptoms at T_2_ can obviously predict the internet addiction at T_3_. Therefore, it will be necessary for further studies to set up more time points and consider classifying subjects to see if there is a difference in the probability of internet addiction among people with different levels of depressive symptoms.

## 5. Conclusions

Ultilizing the RI-CLPM to disaggregate the within-person and between-person effects, our findings demonstrated that depressive symptoms predicted internet addiction but not vice versa at a within-person level. We also found gender differences in the relationship between internet addiction and depressive symptoms; the predictive effect of depressive symptoms on internet addiction was more significant in male group, so more importance should be attached to males. Positive coping strategies had a significant negative predictive effect on the random intercept of internet addiction and depressive symptoms, while negative coping style had a significant positive predictive effect on the random intercept of internet addiction and depressive symptoms. According to the results of our research, adolescents who display a high level of depressive symptoms are in special need of attention, especially males. Prevention and intervention strategies for reducing depressive symptoms may also help reduce internet addiction, and we should pay attention to the cultivation of middle-school students’ positive coping strategies.

## Figures and Tables

**Figure 1 ijerph-18-12869-f001:**
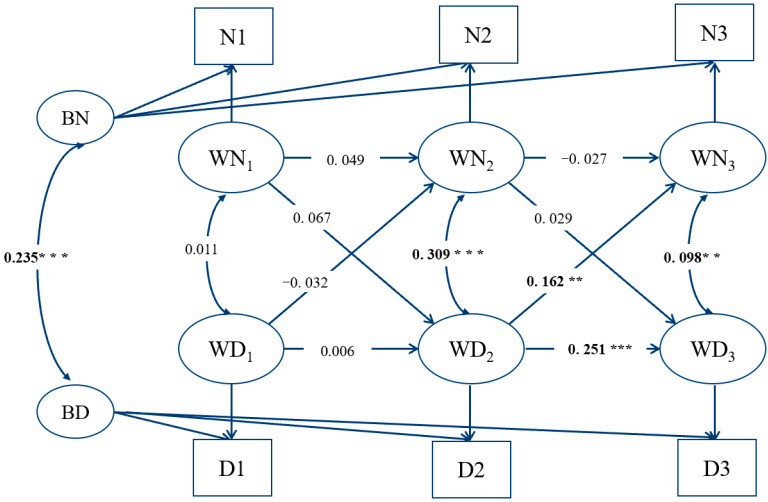
Random intercept cross-lagged model (RI-CLPM) of total sample. Notes. BN = intercept of internet addiction, BD = intercept of depressive symptoms, WN = the within-person fluctuations of internet addiction, WD = the within-person fluctuations of depressive symptoms, N1–N3 are total scores of internet addiction at wave 1–wave 3, D1–D3 are total scores of depressive symptoms at wave 1–wave 3; ** *p* < 0.01, *** *p* < 0.001.

**Figure 2 ijerph-18-12869-f002:**
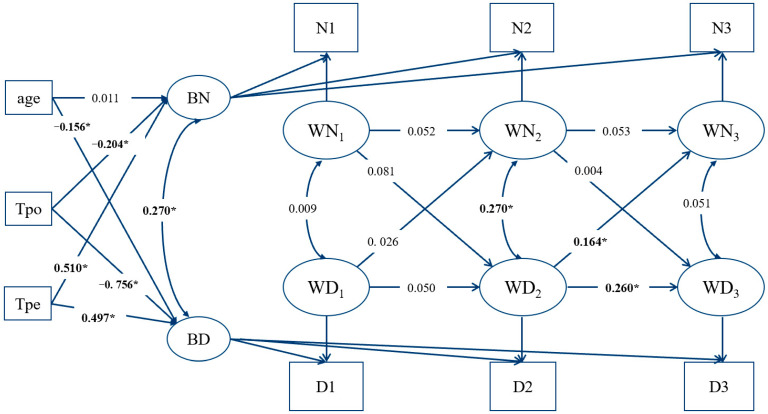
RI-CLPM of male group. Notes. BN = intercept of internet addiction, BD = intercept of depressive symptoms, WN = the within-person fluctuations of internet addiction, WD = the within-person fluctuations of depressive symptoms, N1–N3 are total scores of internet addiction at wave 1–wave 3, D1–D3 are total scores of depressive symptoms at wave 1–wave 3; * *p* < 0.05.

**Figure 3 ijerph-18-12869-f003:**
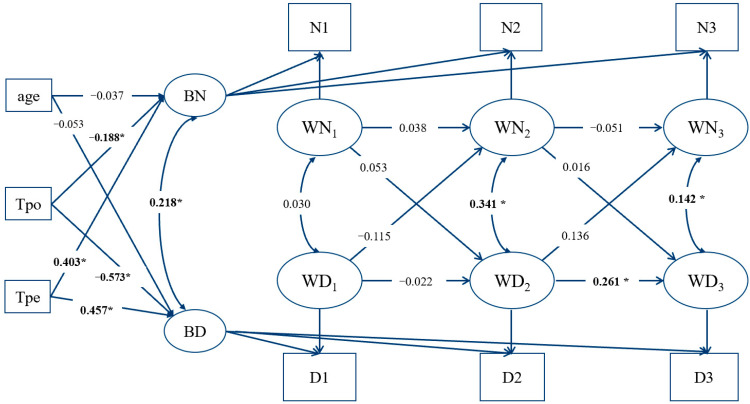
RI-CLPM of female group. Notes. BN = intercept of internet addiction, BD = intercept of depressive symptoms, WN = the within-person fluctuations of internet addiction, WD = the within-person fluctuations of depressive symptoms, N1–N3 are total scores of internet addiction at wave 1–wave 3, D1–D3 are total scores of depressive symptoms at wave 1–wave 3; * *p* < 0.05.

**Table 1 ijerph-18-12869-t001:** Characteristics of participants.

	*N*	%
Gender		
Male	651	44.619
Female	808	55.380
Only child		
yes	820	56.202
no	639	43.797
Age		
11–15	778	53.324
16–19	681	46.675

**Table 2 ijerph-18-12869-t002:** Prevalence of internet addiction (IA) and depressive symptoms.

	First (T_1_)	Second (T_2_)	Third (T_3_)
IA			
Mean (SD)	33.60 (11.91)	32.01 (10.82)	29.72 (10.48)
IA n (%)			
≤50	155 (10.62%)	117 (8.01%)	79 (5.41%)
≤50 < 80	146 (10.00%)	114 (7.81%)	74 (5.07%)
80≤	9 (0.62%)	3 (0.21%)	5 (0.34%)
Depressive symptoms			
Mean (SD)	39.43 (8.42)	37.02 (8.62)	37.21 (9.65)
Depressive symptoms n (%)			
41<	615 (42.15%)	460 (31.53%)	540 (37.01%)

IA: Internet addiction; SD: Standard deviation; T_1_–T_3_: wave 1–wave 3.

**Table 3 ijerph-18-12869-t003:** Correlation coefficients among key variables.

	Variable	IA T_1_	IA T_2_	IA T_3_	DEP T_1_	DEP T_2_	DEP T_3_
1	IA T_1_	1					
2	IA T_2_	0.56	1				
3	IA T_3_	0.57	0.58	1			
4	DEP T_1_	0.24	0.19	0.2	1		
5	DEP T_2_	0.23	0.33	0.24	0.49	1	
6	DEP T_3_	0.20	0.21	0.22	0.44	0.51	1

All correlation coefficients were significant at *p* < 0.001. IA: internet addiction; DEP: depressive symptoms.

**Table 4 ijerph-18-12869-t004:** Model fitting index for male group.

CFI	TLI	SRMR	RMSEA	*χ* ^2^	*p*
0.93	0.82	0.08	0.10	107.18	0.00

**Table 5 ijerph-18-12869-t005:** Model fitting index for female group.

CFI	TLI	SRMR	RMSEA	*χ* ^2^	*p*
0.98	0.95	0.05	0.06	54.46	0.00

## Data Availability

The data are not publicly available due to privacy restrictions.

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
