# Peer review of "The Longitudinal Relationship between Internet Addiction and Depressive Symptoms in Adolescents: A Random-Intercept Cross-Lagged Panel Model"

_ijerph, 2021, doi:10.3390/ijerph182412869_

Round 1

Reviewer 1 Report

There are a lot of spelling errors in the document such as missing letters and no spaces between the sentence conclusions.  These need to be revamped.  

The authors make a conclusion in lines 79-85 about depression in males and females with no citations, proof, or relationships.  This needs to change or back off of the content.  One cannot simply state something that does not have a citation or research to back it up.

Reviewer 2 Report

This study examined the longitudinal relationship between Internet addiction and depression based on a large number of subjects. I have some issues that should be addressed:

(1) This paper requires extensive scientific language editing. Some examples are as follows:

“Although there are several longitudinal researches on the longitudinal…”, Line 47, should be “research”;

“internet” and “Internet” should be unified in the whole manuscript;

Spaces before left parenthesis or after commas and periods cannot be ignored.

(2) Please provide more background information first in Abstract.

(3) The information of Internet users in China in the first paragraph of Introduction should be updated. http://www.cnnic.net.cn/hlwfzyj/hlwxzbg/hlwtjbg/202109/t20210915_71543.htm

(4) The importance of the relationship between Internet addiction and depressive symptoms should be described deeply in Part 1.1, because some other clinical symptoms also significantly relate to Internet addiction (e.g., impulsiveness, compulsivity, anxiety).

(5) “Coping strategy” should be included in Abstract and the second or a new paragraph of Introduction.

(6) In Introduction, the authors said, “so it is necessary to further exam the effect of coping strategies among the relationship between adolescent Internet addiction and depressive symptoms in longitudinal studies.”, but I don’t find these in Results. In fact, coping style just serves a covariate.

(7) In Line 262, “This is the first research using RI-CLPM to …”? The authors have introduced some previous research in Line 50. Else, if all studies are very little different from previous ones, could they all claim to the first? Please delete this statement.

(8) The logic of Discussion should be revised completely. The main doings and findings should be given first, then discuss the main results in points (comparing with previous studies, explaining), and some limitation also should be added lastly.

(9) Why the prediction of depressive symptoms to Internet addiction just occurs in T2 to T3, not other time points? Please explain it.

(10) Conclusions in Conclusion and Abstract parts lack effective information. Even without this study, everyone knows that depressive symptoms should be taken seriously, and reducing depression may reduce Internet addiction.

Round 2

Reviewer 2 Report

Some concerns should be addressed adequately:

(1) My last concern about "Why the prediction of depressive symptoms to Internet addiction just occurs in T2 to T3, not other time points? Please explain it." should not be ignored.

(2) Citation for "Coping strategy, as a relatively stable cognitive and behavioral style, has been proved to be closely related to Internet addiction and depressive symptoms." should be added.

Round 3

Reviewer 2 Report

I don't have more concerns.